# Nutritional Quality of Vegetarian and Non-Vegetarian Dishes at School: Are Nutrient Profiling Systems Sufficiently Informative?

**DOI:** 10.3390/nu12082256

**Published:** 2020-07-28

**Authors:** Romane Poinsot, Florent Vieux, Christophe Dubois, Marlène Perignon, Caroline Méjean, Nicole Darmon

**Affiliations:** 1MOISA, Université de Montpellier, CIRAD, CIHEAM-IAMM, INRAE, Institut Agro, 34060 Montpellier, France; romane.poinsot@ms-nutrition.com (R.P.); marlene.perignon@inrae.fr (M.P.); caroline.mejean@inrae.fr (C.M.); 2MS-Nutrition, Faculté de Médecine La Timone, 13385 Marseille, France; florent.vieux@ms-nutrition.com; 3Trophis, 13170 Les Pennes Mirabeau, France; cdubois.nutrition@free.fr

**Keywords:** children, sustainability, France, school catering, vegetarian, vegan, omega-3 fatty acids, SAIN,LIM, Nutri-Score

## Abstract

In France, school canteens must offer a vegetarian meal at least once per week. The objective was to evaluate the nutritional quality of school main dishes. A database of main dishes served in primary schools was first split into non-vegetarian (*n* = 669) and vegetarian (*n* = 315) categories. The latter has been divided into three sub-categories: vegetarian dishes containing cheese, vegetarian dishes containing eggs and/or dairy products but no cheese and vegetarian dishes without any eggs, cheese or other dairy products (vegan). Categories and sub-categories were compared based on nutrient adequacy ratios for “protective” nutrients (proteins, fibres, vitamins, minerals, essential fatty acids), the contents of nutrients to be limited (saturated fatty acids (SFA), sodium, free sugars) and on two nutrient profiling systems (SAIN,LIM and Nutri-Score). The vegetarian category and the non-vegetarian category displayed “adequate” levels (≥5% adequacy for 100 kcal) on average for almost all “protective” nutrients. The three sub-categories of vegetarian dishes displayed good SAIN,LIM and Nutri-Score profiles on average, although key nutrients were lacking (vitamin B12, vitamin D and DHA) or were present in insufficient amounts (vitamin B2 and calcium) in the vegan sub-category. The sub-category containing eggs and/or dairy products other than cheese was a good compromise, as it provided protective nutrients associated with eggs and fresh dairy products, while the sub-category containing cheese provided higher levels of SFA. Nutrient profile algorithms are insufficiently informative to assess the nutritional quality of school dishes.

## 1. Introduction

School catering is a privileged vector for food education and contributes to facilitating access to healthy and more sustainable food [1]. In France, children are not allowed to bring their own lunch (except for medical reasons) and there is always a set meal proposed to all children. To ensure the good nutritional quality of meals school catering is subject to several laws and recommendations. According to the French regulations [2], a school meal must include four or five components (a protein dish, a side dish, a dairy product, a starter and/or a dessert), and school dishes must comply with service frequencies for fifteen types of dishes in a series of 20 consecutive meals (e.g., side dishes made of vegetables must be served exactly 10 times out of 20) [3]. In addition, in 2019, the introduction of at least one vegetarian meal per week (defined as a meal without meat or fish) became mandatory [4]. To comply with this new law, school canteen managers (dieticians, cooks, etc.) have to adapt their food plan in order to include the weekly vegetarian meal, while still observing the other compulsory service frequencies. At the same time, agri-food industries have anticipated the needs of managers by offering on the market ready-to-eat vegetarian products that can replace meat and fish products, such as soya “steak” or wheat nuggets.

The consumption of meat and seafood facilitates the coverage of requirements for nutrients that are missing or lacking in plant-based products (long-chain omega-3 fatty acids, iodine, vitamins D and B12) and for nutrients such as iron and zinc. Moreover, the bioavailability of iron and zinc is low when they come from plant-based products [5]. However, it is recommended to avoid excessive consumption of red and processed meats to maintain good health [6,7], as they have been associated with an increased risk of cardiovascular disease, type 2 diabetes and colorectal cancer [8]. In the same vein, seafood, and especially fish, are major vectors of contaminants such as methylmercury, arsenic and persistent organic pollutants. Therefore, seafood consumption should be guaranteed to ensure adequate omega-3 fatty acids, vitamin D, selenium and iodine intake but has to be limited to minimize possible detrimental health effects [9]. Plant-based products are rich in fibre, vitamins C and B, magnesium, carotenoids, polyphenols and antioxidants, which contribute to the prevention of chronic diseases [10]. In France, as in most Western countries, the average animal-protein intake is about twice as high as plant-protein intake [11]. In a recent study on a representative French national dietary survey conducted in 2014–2015, simulations on the substitution of animal-protein foods with plant-protein foods in the French diet have shown nutritional benefits, but only when the plant-protein foods were highly diversified (mix of legumes, nuts, seeds and vegetables) [11]. There is clear evidence that a transition towards more plant-based diets will help to reduce diet-related greenhouse gas emissions [12]. However, the total removal of meat is not necessary to ensure nutritional adequacy while reducing the environmental impact of diets [12,13].

Nutrient profiling systems were created to assess the overall nutritional quality of individual foods. They allow foods to be classified, according to their content in some nutrients and other components (e.g., fruit and vegetables) [14]. Two validated nutrient profiling systems were shown to be consistent with French food-based dietary recommendations [15]: the Nutri-Score [16] and SAIN,LIM [17] algorithms. It has been advocated that nutrient profiles could be used by governments in the context of school catering to establish food standards [15] or criteria for food supplies [18].

A previous study on French school meals [19] suggested that the systematic substitution of meat and fish with vegetarian dishes would have a detrimental impact with regard to several indicators of nutritional quality. The data for this previous study had been collected in 2015. However, there were few vegetarian dishes offered by school canteens in France at that time, and they lacked diversity. They mainly contained egg- and/or cheese-based dishes, and, more rarely, dishes made exclusively of plant-based ingredients.

The aim of the present study was to assess the nutritional quality of non-vegetarian main dishes and three sub-categories of vegetarian main dishes currently offered in primary schools in France, based on nutrient adequacy ratios and the Nutri-Score [16] and SAIN,LIM [17] algorithms.

## 2. Material and Methods

The study was initiated and co-developed by the members of the EnScol network (‘’Ensemble, poser les bases d’une évolution des pratiques pour une restauration Scolaire plus durable’’ (“Together, laying the foundations for a shift in practices towards more sustainable school catering”)), a French network of researchers and school catering professionals created in 2019.

### 2.1. Collection of Data

The present study focused on main dishes served to children in primary school (commonly corresponding to 6 to 11 years in France). A main dish can either be a protein dish served with a side dish (e.g., a steak served with green beans) or it can be a complete dish. A complete dish is a dish where the protein “dish” and the side “dish” are incorporated into the same main dish (e.g., lasagne, gratins, chili con carne, etc.) An existing database, based on technical files for all dishes from 40 series of 20 meals, was available thanks to the previous study by Vieux et al. [19] This previous database also contained 40 additional vegetarian dishes (complete dishes, protein dishes and side dishes). All complete dishes and protein dishes were extracted from this database (*n* = 683). The technical files provided the recipes (i.e., the ingredients and quantities used) and, for industrial dishes, the mandatory labelled nutrient contents and the ingredients list.

In 2019, in order to expand the initial database, professionals from the EnScol network were asked to provide technical files for new vegetarian dishes. The latter were provided by four partners: the city of Montpellier; Sodexo, which is a French catering services company; the Association Française des Diététiciens Nutritionnistes (AFDN (French Association of Dieticians, Paris, France)); and the Association pour le Développement de l’Hygiène et de l’Equilibre Nutritionnel (ADHEN (Association for the Development of Health and Nutritional Balance, Saint-Jean-le-Blanc, France)), a French association of dieticians in mass catering.

The ingredient content of the new vegetarian dishes (*n* = 225) provided by EnScol partners was computed following the same procedure applied previously [19]. Their nutritional content was based on CALNUT [20], Nutrinet-Santé [21] or CIQUAL 2013 [22] food composition tables. Duplicate technical files were excluded from the analysis.

### 2.2. Combination of Side Dishes and Protein Dishes to Obtain Main Dishes

Nutritional quality was assessed at the “main dish level”. Complete dishes were thus analysed as such, and protein dishes had to be combined with a side dish. However, technical files of vegetarian protein dishes (except those coming from the 40 series of 20 meals) were not provided with their actual side dish. Therefore, each individual protein dish was combined with an “average” side dish (see Figure 1).

In accordance with the 2011 regulations, vegetable side dishes must be served at the same frequency as starch side dishes (i.e., 10 times out of 20). Two average side dishes (a “vegetables” side dish and a “starches” side dish) were created based on the nutritional content of all the “vegetables” side dishes (containing at least 50% vegetables) and of all the “starches” side dishes (containing at least 50% starch) extracted from the initial 40 series of 20 meals [19]. Their average nutritional content was calculated, weighted by their average recommended portion size, determined by a dietician in the previous study.

Next, the type of side dish usually served with each protein dish was identified. This could be either a “vegetables” side dish, a “starches” side dish, or both if the protein dish was served as often with vegetables or starches. For protein dishes previously collected from the 40 series of meals, the type of side dish was determined directly, according to the composition of the meals in which the dishes were present. For example, if a protein dish was served with rice, the average side dish corresponding to this protein dish was “starches”. If, in another meal, the protein dish was served with green beans, it was combined with both a “starches” and a “vegetables” side dish and the two options were calculated separately. For the vegetarian protein dishes collected, their combination with side dishes was determined according to the expertise of EnScol partners: vegetables, starches or both.

The minimum recommended portion size of the protein source in a main dish is 70 g for all children in primary schools [3]. On average, the recommended portion sizes in the 40 series of meals were 113 g for “vegetables” side dishes and 170 g for “starches” side dishes. These portion sizes were used for the associations of each protein dish (70 g) with its corresponding average side dish(es) (113 g for vegetables and 170 g for starches). The nutritional content of each main dish was calculated for 100 g of the final dish. Below, the term “main dish” is indifferently shortened to “dish” or kept as “main dish”.

### 2.3. Categorisation of Vegetarian Dishes

Main dishes were split into two categories: vegetarian and non-vegetarian dishes. In order to establish a typology of vegetarian dishes, they were then divided into 3 different sub-categories according to the presence or absence of cheese, eggs and dairy products other than cheese (milk, yoghurt, butter, etc.) The amounts of eggs, cheese and dairy products (DP) other than cheese were estimated from recipes:All the vegetarian dishes containing cheese were included in the sub-category “CHEESE (and/or other DP and/or EGG”. Dishes from this sub-category could also contain dairy products other than cheese and eggs.All the vegetarian dishes containing eggs and/or dairy products but no cheese were included in the sub-category “EGG and/or DP (excl. CHEESE)”.All the vegetarian dishes without any eggs, cheese or other dairy products were included in the sub-category “VEGAN”.

### 2.4. Evaluation of the Nutritional Quality of Main Dishes

#### 2.4.1. Nutritional Content of Main Dishes

Energy, fibres, macro- and micronutrient contents were assessed for each main dish in each category and sub-category. Nutrient adequacy ratios were calculated for each “protective” nutrient in each dish, as the percentage of the daily recommended value per 100 kcal of main dish. The so-called “protective” nutrients are protein, fibres vitamins B1, B2, B6, B9, B12, C, D, E and A, calcium, potassium, iron, magnesium, zinc, copper, iodine, selenium, linoleic acid (LA), alpha-linolenic acid (ALA) and docosahexaenoic acid (DHA). The recommended values used in the computation of nutrient adequacy ratios are those in Table A1. Daily recommended values have been calculated taking into account the age and sex distribution of primary school children in France, according to a previously described methodology [19]. The level of each nutrient was considered “adequate” if the median nutrient adequacy ratio was greater than or equal to 5%/100 kcal. A threshold of 5% was used because 5% adequacy for 100 kcal is equivalent to 100% for 2000 kcal (considered a reference energy intake). Contents in nutrients to be limited and energy density were calculated for 100 g of dish. The so-called nutrients “to be limited” are sodium, saturated fatty acids (SFA), total sugars and free sugars [23].

#### 2.4.2. Nutri-Score System

The Nutri-Score algorithm is a five-colour nutrition label derived from the Nutrient Profiling System of the British Food Standards Agency (modified version) [24]. Nutri-Score is based on a positive sub-score P and a negative sub-score N [16]. The P sub-score includes protein and fibre contents in 100 g of product and the percentage of “fruits, vegetables, legumes, nuts as well as (since 2019 [25]) colza, walnuts and olive oils” incorporated in the recipe. The N sub-score includes energy, total sugars, sodium and SFA per 100 g of final product. The final score, obtained by subtracting P from N, is submitted to 4 thresholds defining 5 classes. The five Nutri-Score classes are associated with 5 colours and 5 letters, from dark green (A) for “good nutritional quality” to dark orange (E) for “lower nutritional quality”. Each main dish from the database was allocated to one of the 5 possible Nutri-Score classes.

#### 2.4.3. SAIN,LIM System

The SAIN,LIM algorithm is based on the SAIN (score of nutritional adequacy of individual foods) and the LIM (score of nutrients to be limited) sub-scores, which reflect the favourable and unfavourable aspects of foods, respectively [17]. The SAIN score is an unweighted arithmetic mean of the percentage adequacy for five positive nutrients (plus one optional nutrient): proteins, fibres, calcium, iron and vitamin C (plus vitamin D) in 100 kcal of final food. The LIM score is the mean percentage of the maximal recommended values (MRV) for three nutrients to be limited in a healthy diet: sodium, SFA and free sugars in 100 g of final food. The threshold for the SAIN is 5, corresponding to 5% adequacy for 100 kcal, i.e., 100% for a reference daily energy intake of 2000 kcal. The threshold for the LIM is 7.5, corresponding to 7.5% adequacy for 100 g, i.e., 100% for a reference daily intake quantity of 1333 g. Attributing a threshold to each score enables the definition of four classes: class 1 (the most favourable profile), SAIN ≥ 5 and LIM < 7.5; class 2, SAIN < 5 and LIM < 7.5; class 3, SAIN ≥ 5 and LIM ≥ 7.5; and class 4, SAIN< 5 and LIM > 7.5 (the least favourable profile). Where enough information was available, each main dish from the database was allocated to 1 of the 4 possible SAIN,LIM classes.

### 2.5. Statistical Analysis

Quantitative variables (e.g., nutrient adequacy ratios) were compared between categories (vegetarian and non-vegetarian) using the Wilcoxon test and between the three vegetarian sub-categories using the Kruskal–Wallis test. Where the difference between the three sub-categories was significant, a post hoc comparison (Dunn’s test) was performed. Nutrient adequacy ratios were also compared to the theoretical 5% threshold using the Wilcoxon-Mann-Whitney test for one sample. Where the median and the mean were not convergent, a one-sided test was used to determine whether the nutrient level was significantly lower or higher than 5%. The distribution of dishes in the Nutri-Score and SAIN,LIM classes between the two categories, and then between the three sub-categories, was compared with chi-squared tests of χ^2^. Where the number of dishes per class did not allow the test to be carried out, class grouping was performed. Agreement between Nutri-Score and SAIN,LIM was measured with Cohen’s kappa test. Correlations between total Nutri-Score and LIM and between total Nutri-Score and SAIN were measured with Spearman’s coefficient test. Finally, the probability of classification in the best nutrient profiling class according to the category of main dish or the sub-category of vegetarian dish was assessed by binary logistic regression adjusted for the production method (industrial or not) and the type of side dish (“vegetables”, “starches” or “none”).

Statistical analyses were performed with R software version 3.6. The level of significance was set to 5% for all of the tests.

## 3. Results

### 3.1. Database Constitution

The city of Montpellier provided technical files of locally prepared (*n* = 3) and industrial (*n* = 11) vegetarian dishes that were offered at least once in the public schools of Montpellier. Sodexo provided all technical files for all of the non-industrial vegetarian dishes (*n* = 79) in their database in April 2019 and for industrial vegetarian products (*n* = 13) presented by suppliers but not necessarily served in school canteens run by the company. The AFDN provided a list of 75 industrial technical files, originating from a regular inventory of new industrial vegetarian products offered by manufacturers to French school caterers from January to December 2018 [26], as well as 5 technical files for non-industrial vegetarian dishes served in the canteen of a small town in western France. The ADHEN extracted non-industrial technical files for vegetarian dishes (*n* = 39) from their “Menu-Co” database software, excluding incomplete files and the files that had already been provided for the previous study by Vieux et al. [19].

Overall, 225 new technical files were collected from professional members of EnScol in 2019. When added to protein dishes and complete dishes from the previous study, a total of 267 vegetarian dishes and 669 non-vegetarian dishes were counted (Figure 1). A total of 73 non-vegetarian and 133 vegetarian main dishes were complete dishes. After the association of protein dishes with their corresponding average side dish(es), 277 non-vegetarian and 128 vegetarian main dishes composed of one protein dish and the average “vegetables” side dish and 319 non-vegetarian and 92 vegetarian main dishes composed of one protein dish and the average “starches” side dish were obtained. It was not possible to calculate the SAIN,LIM profile of 38 industrial vegetarian dishes, owing to a lack of sufficient information in the original data provided. The final database contained a total number of 984 main dishes, including 669 non-vegetarian dishes and 315 vegetarian dishes, the latter containing 129 “CHEESE and/or DP and/or EGG” dishes, 53 “EGG and/or DP (excl. CHEESE)” dishes and 133 “VEGAN” dishes.

### 3.2. Comparisons Between Non-Vegetarian and Vegetarian Dishes

Non-vegetarian and vegetarian dishes had the same energy density (Table 1). Most “protective” nutrients were present in “adequate” amounts (i.e., adequacy ratio not lower than 5%) on average in each of the two categories of main dishes. For those nutrients present in “adequate” amounts, their adequacy ratios were either: not significantly different between the two categories (vitamins B1, C and E, iodine and selenium); higher in non-vegetarian dishes (vitamins B2, B3, B6 and B12, potassium and zinc); or higher in vegetarian dishes (fibres, vitamins B9 and A, iron, magnesium, copper and LA). For proteins, their nutrient adequacy ratios were well above the 5% threshold: 24.9% and 17.2% on average for non-vegetarian and vegetarian dishes, respectively. The average calcium content was “adequate” in vegetarian dishes but not in non-vegetarian dishes. Vitamin D and DHA were present at insufficient levels in both categories, and the levels were significantly lower in vegetarian dishes than in non-vegetarian dishes. ALA was also present at insufficient levels in both categories, and the average level was significantly lower in non-vegetarian dishes than in the vegetarian dishes.

The contents of SFA and sodium were in the same order of magnitude between vegetarian and non-vegetarian dishes, yet slightly but significantly lower in vegetarian dishes. The content of free sugars was negligible in both categories of dishes.

Whatever the category, most dishes were classified in classes A and B of the Nutri-Score, and in classes 1 and 2 of the SAIN,LIM. No dishes were classified in the least favourable class of the Nutri-Score (i.e., class E) and very few were in class D (0.9% and 1.0% of non-vegetarian and vegetarian dishes, respectively). In contrast, 7.6% of non-vegetarian dishes and 4.8% of vegetarian dishes were in the least favourable class of the SAIN,LIM, i.e., class 4 (Figure 2). Given that the number of dishes per class was sometimes too low to perform χ^2^ tests, Nutri-Score classes B, C and D were grouped, as well as classes 2, 3 and 4 of the SAIN,LIM, and they were compared to the respective best classes (i.e., Nutri-Score class A and SAIN,LIM class 1). The proportion of dishes classified in the best class of the SAIN,LIM (class 1) was significantly higher for vegetarian dishes than for non-vegetarian dishes (65.7% vs. 40.4%, *p* < 0.001). With the Nutri-Score, the difference between the two categories followed the same trend as with the SAIN,LIM, but was smaller (66.2% and 59.2% of vegetarian and non-vegetarian dishes were in class A, respectively, *p* = 0.029).

### 3.3. Comparisons between the Three Sub-Categories of Vegetarian Dishes

On average, nutrient contents were “adequate” in all three sub-categories of vegetarian dishes for fibres, vitamins B1, B3, B6, B9, A and E, potassium, iron, magnesium, copper, iodine, selenium and LA (Table 2). The “VEGAN” sub-category had the lowest SFA content and the highest adequacy ratios for fibres, vitamin B3 and copper. However, vitamin B2, calcium and ALA were present at insufficient levels and vitamins B12 and D and DHA were almost absent (less than 1%/100 kcal) in the “VEGAN” sub-category. The two other sub-categories had adequate contents of calcium and vitamins B2 and B12, and higher vitamin D and DHA contents than the “VEGAN” sub-category. On average, the “EGG and/or DP (excl. CHEESE)” category had adequate levels of ALA and DHA. The “CHEESE (and/or other DP and/or EGG)” category had the highest calcium content, but contained more SFA than the other two categories.

With the Nutri-Score, “VEGAN” dishes were more often classified in class A than “EGG and/or DP (excl. CHEESE)” dishes (86.5% vs. 64.2%, *p* = 0.001) and than “CHEESE (and/or other DP and/or EGG)” dishes (86.5% vs. 47.3%, *p* < 0,001). The difference between the “EGG and/or DP (excl. CHEESE)” category and the “CHEESE (and/or other DP and/or EGG)” category did not reach significance (*p* = 0.057) (Figure 3). With the SAIN,LIM, “VEGAN” dishes were more often classified in class 1 than “CHEESE (and/or other DP and/or EGG)” dishes (70.7% vs. 55.8%, *p* = 0.018), but the proportion of dishes in class 1 was not significantly different between the “VEGAN” and the “EGG and/or DP (excl. CHEESE)” sub-categories (70.7% vs. 77.4%, *p* = 0.459).

## 4. Discussion

Vegetarian and non-vegetarian main dishes served to children in primary schools in France generally showed good SAIN,LIM and Nutri-Score profiles on average, despite low omega-3 fatty acids and vitamin D levels. Among the vegetarian dishes, the vegan dishes (i.e., without any ingredients of animal origin) had the lowest SFA and the highest fibre contents of the three sub-categories, and they displayed good Nutri-Score and SAIN,LIM profiles, but this was hiding inadequate levels of vitamins B2 and B12 and calcium as well as the lack of vitamin D and DHA. Vegetarian dishes containing eggs and/or dairy products other than cheese showed a good nutritional compromise between vegan dishes and cheese-containing vegetarian dishes because they provided the “protective” nutrients associated with eggs and fresh dairy products (calcium, vitamins B2, B12 and D) without cheese-associated SFA.

To the best of our knowledge, this was the first time that the nutritional quality of vegetarian dishes in schools has been assessed for a wide spectrum of dishes. A Belgian study [27] compared five meat components of conventional main courses and five vegetarian components of main courses served for “Thursday Veggie Day” in primary schools in Belgium, and it only focused on macronutrients and fibres. In accordance with our results, it found that meat components were lower in fibres than vegetarian components and that a specific vegetarian component containing cheese sauce was too high in SFA.

The agreement between the SAIN,LIM and Nutri-Score was low (κ = 0.45) and only 60.9% of the dishes were similarly classified with both systems (Appendix C). In particular, among the vegetarian sub-categories, the vegan sub-category was the best according to the Nutri-Score system, but not according to the SAIN,LIM system (which similarly classified dishes from the two sub-categories of vegetarian dishes without cheese). In addition, neither the Nutri-Score nor the SAIN,LIM were able to detect the nutritional defects of the vegan dishes. The inadequate contents in vitamins B2, B12 and D, calcium and DHA in vegan dishes were not captured by the nutrient profiling assessments. Therefore, although it is claimed that nutrient profiling systems are helpful to assess the nutritional quality of foods at first sight, summarising nutritional information in a single score or grade can lead to misinterpretations [28]. For example, in a study conducted on Dutch products, the Nutri-Score system failed to properly classify all ready meals, more than half of them being classified in class A or B even when they contained high values of salt [29]. The present study suggests that the SAIN,LIM and the Nutri-Score are insufficiently informative to be used alone as reliable criteria for choosing school food supplies or establishing school dietary guidelines.

The protein contents were well above the 5%/100 kcal threshold for the three sub-categories of vegetarian dishes. In fact, protein deficiency is almost impossible with regard to children at primary school, even with vegetarian or vegan dishes, since the protein recommendation for children is relatively low compared to their energy needs [30]. The level of amino acids has not been estimated, but given that protein content is high, it can be assumed that amino acid requirements are easily met by vegetarian dishes, despite a less optimal distribution profile of amino acids in plant-based foods [30]. Iron was present at an “adequate” level in vegetarian dishes, but with a necessarily lower content of haem iron (only present in animal-based foods, better absorbed and better available than non-haem iron) and, a higher content of non-haem iron than in non-vegetarian dishes. However, assuming that the respective absorptions of iron and zinc from vegetarian dishes are only 5% and 15%, respectively (instead of 10% and 30% in an omnivorous diet, the respective assumption percentages used to define iron and zinc recommendations [31]), this would double the biological requirements for iron and zinc and, therefore, halve the percentage coverage in 100 kcal. In this respect, it should be noted that the bioavailability of iron (especially non-haem iron) and zinc is altered by the presence of phytates, which are present in certain fibre-rich plants such as whole grains and pulses [31,32]. The very high level of fibres in vegan dishes should therefore be given special attention. In children, an excessive intake of fibres may reduce energy intake and cause nutrient deficiencies [33].

There are essential nutrients that are critical to children’s health but that are not included in the computation of the nutrient profiling systems. Vitamin D and DHA levels were low in all sub-categories of vegetarian dishes and almost absent in vegan dishes, because these two soluble nutrients are mostly provided by fatty fish and, to a lesser extent, by eggs. This must remain a point of vigilance, given that the French diet is already deficient in these two nutrients [34]. Moreover, the calcium level was below 5% in vegan dishes but the highest (i.e., 6.8%) in vegetarian dishes containing cheese. Dairy products are important contributors of calcium, iodine and vitamins D and B12 for French children aged 1 to 10 years [34], however, between 2010 and 2016, the proportion of French children aged 6 to 10 years meeting the calcium recommendation decreased from 67% to 55% [35]. It is therefore important that school meals provide sufficient amounts of calcium.

In 2016, a position paper of the American Dietetic Association (ADA) stated that “appropriately planned vegetarian diets, including vegan diets, are healthful, nutritionally adequate, and may provide health benefits in the prevention and treatment of certain diseases. These diets are appropriate for all stages of the life cycle, including […] childhood” [36]. However, at the same time, the paper acknowledged that for several key nutrients, such as DHA, vitamin D, vitamin B12, calcium, zinc, iron and iodine, meeting the requirements may be challenging with exclusively plant-based diets, and that recourse to fortified foods and supplementation may prove useful, if not necessary. Nonetheless, potential associations between serious risks in terms of brain and body development in children and vegetarian diets were not discussed by the ADA [37]. In Europe, the German Society for Pediatric and Adolescent Medicine does not encourage vegetarian diets in childhood without paediatric supervision and nutrient supplementation [38]. In France and Belgium, medical societies are more radical with regard to vegan diets [39,40], arguing that they would expose children to severe nutritional deficiencies at a key period of their development [39]. Vegetarian diets in adults are associated with protective effects, such as a reduction in the total cancer incidence and the incidence and/or mortality of ischemic heart disease [41], but existing studies do not provide conclusive evidence to decisively declare that “appropriately planned” vegetarian diets are as healthy as “appropriately planned” omnivorous diets [37,42,43].

Environmental and health aspects of food are strongly linked [44]. In French primary schools, main dishes were found to account for 60% of total meal waste, while being the greatest contributors to the cost and environmental impact of the meal [45]. Therefore, the replacement of main dishes (and more particularly the protein dish containing fish or meat) appears to be a favourable option to improve the sustainability of school lunches. Studies showed that it is possible to decrease the carbon [46,47,48,49,50] and water [47] impacts of school meals while maintaining their nutritional adequacy by combining all categories of food including meat and fish, showing that diversified and omnivorous meals have a full place in sustainable school lunches. However, more work is needed to determine the optimal frequency of vegetarian dishes and non-vegetarian dishes in school catering. Furthermore, the multicriteria optimisation of existing dishes to simultaneously improve their nutritional content and their environmental impact under cost constraints is a promising approach to developing more sustainable school meals [50].

One strength of this study is its use of a unique database of a considerable number of main dishes usually served in primary schools [19], completed with recent vegetarian dishes specifically collected for this study. Except for a few industrial dishes, recipes were based on specific information, and not on standard recipes or generic products. There are limitations to this work, which provide a roadmap for future work. First of all, the database of main dishes used for this study cannot claim to be representative of all dishes served by school caterers in France. The database is not homogeneous since technical files of non-vegetarian and most of the vegetarian dishes were obtained in 2015 and 2019, respectively. Nevertheless, both sets of technical files were collected and analysed following the same methodology. In addition, they reflect current practices, as they were provided by the major stakeholders in the sector. Another limitation is that the average side dishes constituted more than half of most of the main dishes studied and were only divided into two types: “starches” or “vegetables”. Moreover, main dishes could be industrially produced or not. Binary logistic regressions with adjustment for the type of side dish and the production method (industrial or not) were performed. They confirmed that the probabilities of classification in Nutri-Score class A or SAIN,LIM class 1 were significantly higher for vegetarian dishes than for non-vegetarian dishes (Appendix A). Similarly, the differences between the three sub-categories of vegetarian dishes (Appendix A) were still significant with adjustments. Dishes served with a “vegetables” average side dish were more likely to be classified in class A and class 1 than those served with a “starches” average side dish. In terms of practical implications, this indicates that it would be necessary to improve the overall nutritional quality of “starches” side dishes, perhaps using refined cereals less often, and whole grains and legumes more often, but this requires specific exploration that could not be carried out within the framework of this study. The bioavailability of nutrients was not considered in the nutritional assessment. To properly consider the bioavailability of nutrients, it would be more relevant to assess each dish as part of an entire meal, and more broadly as part of a series of meals.

## 5. Conclusions

The overall nutritional quality of the main dishes offered in primary schools in France, whether vegetarian or not, was generally good based on the nutrient profiling system assessment. However, in all categories and sub-categories of main dishes, omega-3 fatty acids and vitamin D were present in inadequate amounts on average, especially in the vegan sub-category. Main dishes from the cheese-containing vegetarian sub-category were too high in SFA and more often classified in the least favourable Nutri-Score and SAIN,LIM classes than main dishes from the two other vegetarian sub-categories. Neither the Nutri-Score, nor the SAIN,LIM systems were able to detect the insufficient amounts of calcium, vitamin B2 and ALA, and the lack of vitamin B12, vitamin D and DHA in the sub-category of vegan dishes.

The present study demonstrates that nutrient profiling systems algorithms were not sufficient to determine the nutritional adequacy of school dishes, since they did not cover all “protective” nutrients. It is therefore advisable to maintain a critical approach towards these algorithms with the aim to ensure adequate nutrition for children.

## Figures and Tables

**Figure 1 nutrients-12-02256-f001:**
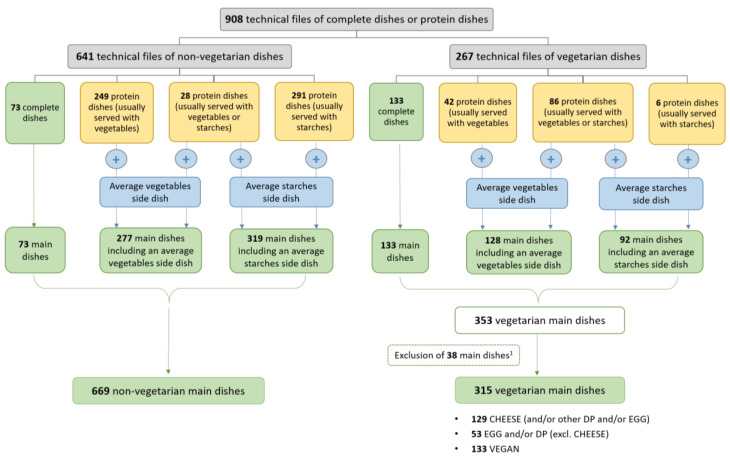
Overview of the distribution of main dishes according to their category (vegetarian or non-vegetarian), the type of side dish (“no side dish”, “vegetables” or “starches”) and, for vegetarian dishes, the sub-category (“CHEESE and/or DP and/or EGG”, “EGG and/or DP (excl. CHEESE)”, “VEGAN”). ^1^ SAIN and LIM scores were not estimated for 38 industrial main dishes because their technical file was incomplete, labelled information on nutrients was insufficient and no matching food was found in the ingredient database.

**Figure 2 nutrients-12-02256-f002:**
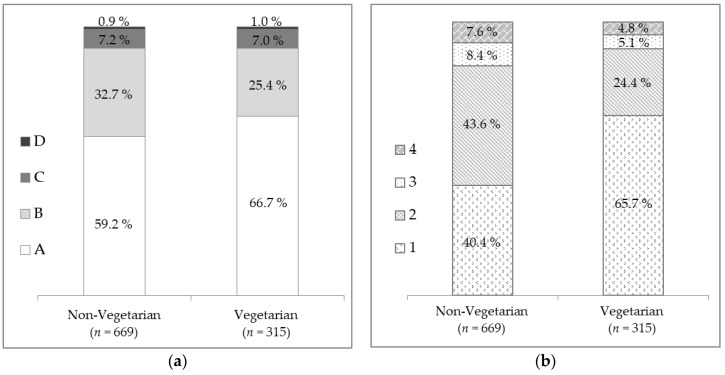
Nutrient profile of non-vegetarian (*n* = 669) and vegetarian (*n* = 315) main dishes according to (**a**) Nutri-Score classes (A, B, C and D); (**b**) SAIN,LIM classes (1, 2, 3 and 4).

**Figure 3 nutrients-12-02256-f003:**
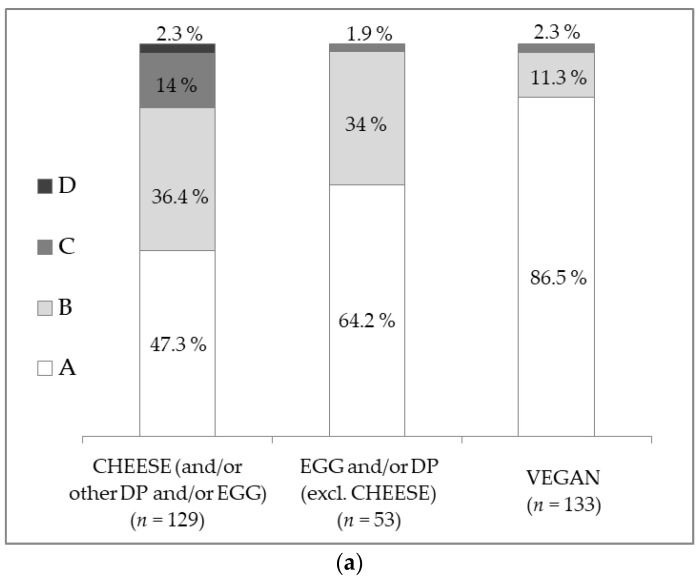
Nutrient profile of the three sub-categories of vegetarian main dishes: “CHEESE (and/or other DP and/or EGG)” (*n* = 129), “EGG and/or DP (excl. CHEESE)” (*n* = 53) and “VEGAN” (*n* = 133), according to (**a**) Nutri-Score classes (A, B, C and D); (**b**) SAIN,LIM classes (1, 2, 3 and 4).

**Table 1 nutrients-12-02256-t001:** Energy density and other nutritional characteristics of non-vegetarian (*n* = 669) and vegetarian (n = 315) main dishes (Med = Median; M = Mean; SD = Standard Deviation).

	Non-Vegetarian Main Dishes (n = 669)			Vegetarian Main Dishes (n = 315)			
	Med	M	SD	Med	M	SD	*p* ^1^
Energy density (kcal/100 g)	136	134	33.0	128	133	38.1	0.160
Carbohydrates (g/100 g)	17.7	12.9	6.97	13.3	14.8	7.66	<0.001
Fats (g/100 g)	4.76	5.22	2.13	4.78	5.03	2.19	0.619
Proteins (g/100 g)	7.80	7.98	1.98	5.51	5.65	1.94	<0.001
Proteins (%/100 kcal) ^2^	22.8	24.9	8.14	16.7	17.4	5.49	<0.001
Fibres (%/100 kcal) ^2^	9.29	10.3	3.44	15.1	16.9	9.27	<0.001
Vitamin B1 (%/100 kcal) ^2^	8.10	9.54	5.26	8.06	8.76	3.88	0.298
Vitamin B2 (%/100 kcal) ^2^	5.00	5.57	3.28	4.76 ^3^	5.49	3.63	0.017
Vitamin B3 (%/100 kcal) ^2^	12.9	16.3	9.68	5.53	5.85	2.47	<0.001
Vitamin B6 (%/100 kcal) ^2^	10.9	11.9	4.85	7.50	8.19	3.61	<0.001
Vitamin B9 (%/100 kcal) ^2^	6.21	10.1	6.44	15.2	16.7	10.7	<0.001
Vitamin B12 (%/100 kcal) ^2^	16.7	22.3	19.0	3.36 ^3^	6.51	7.85	<0.001
Vitamin C (%/100 kcal) ^2^	3.48 ^4^	5.70	4.17	4.36 ^3^	5.51	4.58	0.136
Vitamin D (%/100 kcal) ^2^	**1.67 ^5^**	3.65	6.16	**0.71**	1.66	2.30	<0.001
Vitamin E (%/100 kcal) ^2^	7.36	8.83	5.31	7.92	9.47	6.52	0.674
Vitamin A (%/100 kcal) ^2^	5.18	14.4	13.4	9.13	16.9	20.0	0.026
Calcium (%/100 kcal) ^2^	**2.33**	2.96	1.83	4.84 ^3^	5.52	3.31	<0.001
Potassium (%/100 kcal) ^2^	5.77	6.39	2.01	5.56	6.04	2.69	0.001
Iron (%/100 kcal) ^2^	7.33	8.01	3.72	9.59	10.4	5.34	<0.001
Magnesium (%/100 kcal) ^2^	8.28	8.82	2.46	10.4	12.3	6.75	<0.001
Zinc (%/100 kcal) ^2^	5.75	7.43	5.12	5.40	5.70	1.99	0.008
Copper (%/100 kcal) ^2^	6.62	6.68	1.76	8.43	9.04	4.15	<0.001
Iodine (%/100 kcal) ^2^	6.10	8.66	7.75	6.37	8.20	8.30	0.862
Selenium (%/100 kcal) ^2^	8.83	11.4	8.26	8.54	10.1	5.50	0.191
LA ^7^ (%/100 kcal) ^2^	5.26	5.98	3.13	5.85	7.48	5.47	0.017
ALA ^8^ (%/100 kcal) ^2^	**2.12**	3.01	2.10	**3.22**	4.10	3.38	<0.001
DHA ^9^ (%/100 kcal) ^2^	**0.62 ^6^**	13.1	29.5	**0.21**	1.67	3.68	<0.001
SFA ^10^ (g/100 g)	1.50	1.8	1.01	1.21	1.60	1.16	<0.001
Sodium (mg/100 g)	259	287	101	249	258	190	<0.001
Total sugars (g/100 g)	1.20	1.3	0.75	1.85	1.95	1.11	<0.001
Free sugars (g/100 g)	0.07	0.109	0.2	0.07	0.115	0.35	0.005

^1^ Wilcoxon–Mann–Whitney test. ^2^ Nutrient adequacy ratio of recommended intake for children aged 4–13 years attending primary school [19], expressed per 100 kcal of dish. ^3^ Not significantly different from 5%/100 kcal. ^4^ One-sided test: significantly higher than 5%. ^5^ All nutrient adequacy ratios significantly lower than 5%/100 kcal are shown in **
bold**. ^6^ One-sided test: significantly lower than 5%. ^7^ Linoleic acid. ^8^ Alpha-linolenic acid. ^9^ Docosahexaenoic acid. ^10^ Saturated fatty acids.

**Table 2 nutrients-12-02256-t002:** Energy density and other nutritional characteristics of the three sub-categories of vegetarian main dishes: CHEESE (and/or other DP and/or EGG) (*n* = 129), EGG and/or DP (excl. CHEESE) (*n* = 53) and VEGAN *(n* = 133) (Med = Median; M = Mean; SD = Standard Deviation).

	CHEESE (and/or Other DP and/or EGG) (n = 129)			EGG and/or DP (excl. CHEESE) (n = 53)			VEGAN (n = 133)			
	Med	M	SD	Med	M	SD	Med	M	SD	*p* ^1^
Energy density (kcal/100 g)	130 ^a,2^	136	39.0	116 ^b^	116	33.5	133 ^a^	136	37.5	0.003
Carbohydrates (g/100 g)	12.3 ^a^	13.9	6.81	9.86 ^b^	10.	6.04	18.0 ^c^	17.1	8.23	<0.001
Fats (g/100 g)	5.33 ^a^	5.83	2.39	4.86 ^a^	5.12	2.07	4.40 ^b^	4.23	1.69	<0.001
Proteins (g/100 g)	5.87	5.89	1.86	5.39	5.27	1.67	5.08	5.56	2.08	0.148
Proteins (%/100 kcal) ^3^	16.9 ^a^	17.6	4.39	17.9 ^a^	18.5	5.52	15.3 ^b^	16.7	6.33	0.008
Fibres (%/100 kcal) ^3^	12.3 ^a^	12.9	6.38	14.6 ^a^	16.3	8.03	18.4 ^b^	20.7	10.3	<0.001
Vitamin B1 (%/100 kcal) ^3^	6.87 ^a^	7.57	3.56	8.06 ^b^	9.37	4.22	8.90 ^b^	9.67	3.76	<0.001
Vitamin B2 (%/100 kcal) ^3^	5.45 ^a^	6.18	3.42	7.57 ^a^	8.04	5.12	**3.31 ^5,b^**	3.80	1.91	<0.001
Vitamin B3 (%/100 kcal) ^3^	4.76 ^4,a^	5.22	2.52	5.35 ^a,b^	5.66	1.84	6.25^b^	6.53	2.48	<0.001
Vitamin B6 (%/100 kcal) ^3^	6.96 ^a^	7.25	3.12	8.34 ^b^	9.22	4.24	8.05 ^b^	8.71	3.59	<0.001
Vitamin B9 (%/100 kcal) ^3^	11.8 ^a^	13.5	7.90	18.3 ^b^	20.7	12.2	16.6 ^b^	18.1	11.6	<0.001
Vitamin B12 (%/100 kcal) ^3^	7.76 ^a^	10.0	7.77	6.67 ^a^	9.85	10.2	**0.71 ^b^**	1.75	2.69	<0.001
Vitamin C (%/100 kcal) ^3^	4.36 ^4^	5.12	3.94	6.08 ^4^	6.62	5.30	3.52 ^4^	5.45	4.81	0.238
Vitamin D (%/100 kcal) ^3^	**1.38 ^a^**	2.11	2.21	**1.97 ^a^**	3.38	3.49	**0.40 ^b^**	0.53	0.63	<0.001
Vitamin E (%/100 kcal) ^3^	6.97 ^a^	8.61	6.24	11 ^b^	11.8	7.41	7.55 ^a,b^	9.36	6.23	0.010
Vitamin A (%/100 kcal) ^3^	8.99 ^a^	16.3	15.4	19.3 ^a^	24.5	32.0	6.42 ^b^	14.5	17.1	0.001
Calcium (%/100 kcal) ^3^	6.76 ^a^	7.46	3.61	5.04 ^4,b^	5.25	2.49	**3.21 ^c^**	3.74	2.00	<0.001
Potassium (%/100 kcal) ^3^	4.82 ^4,a^	5.19	2.34	6.04 ^b^	6.90	3.04	5.99 ^b^	6.53	2.64	<0.001
Iron (%/100 kcal) ^3^	7.64 ^a^	8.54	4.95	11.0 ^b^	11.6	5.04	10.4 ^b^	11.8	5.30	<0.001
Magnesium (%/100 kcal) ^3^	8.26 ^a^	10.2	5.80	10.4 ^b^	12.2	5.69	12.8 ^b^	14.3	7.43	<0.001
Zinc (%/100 kcal) ^3^	5.72	5.93	1.91	5.30 ^4^	5.58	1.87	4.98 ^6^	5.54	2.11	0.122
Copper (%/100 kcal) ^3^	6.87 ^a^	7.36	3.54	7.85 ^b^	9.24	4.90	10.1 ^c^	10.6	3.75	<0.001
Iodine (%/100 kcal) ^3^	7.07 ^a^	9.19	11.5	10.4 ^b^	11.0	6.87	5.74 ^c^	6.13	2.92	<0.001
Selenium (%/100 kcal) ^3^	8.55	9.82	4.82	9.54	10.7	4.83	8.08	10.2	6.34	0.454
LA (%/100 kcal) ^3^	4.59 ^4,a^	6.20	5.14	9.47 ^b^	9.74	5.59	6.41 ^c^	7.83	5.44	<0.001
ALA (%/100 kcal) ^3^	**2.60 ^a^**	3.33	2.38	4.44 ^4,b^	4.45	2.24	**3.21 ^b^**	4.69	4.33	0.009
DHA (%/100 kcal) ^3^	**0.42 ^a^**	1.75	3.25	1.16 ^4,a^	4.83	6.30	**0.01 ^b^**	0.33	0.71	<0.001
SFA (g/100 g)	1.94 ^a^	2.35	1.37	1.23 ^b^	1.37	0.59	0.92 ^c^	0.87	0.44	<0.001
Sodium (mg/100 g)	249	275	268	251	254	85.0	248	244	98.3	0.394
Total sugars (g/100 g)	1.93	1.96	1.20	1.98	2.06	1.03	1.74	1.91	1.06	0.526
Free sugars (g/100 g)	0.05	0.12	0.43	0.07	0.15	0.39	0.04	0.10	0.21	0.274

^1^ Wilcoxon–Mann–Whitney test. ^2^ Same index letters (e.g. a and a) indicates that there is no significant difference between the two sub-categories and different index letters (e.g. a and b) indicate that the difference is statistically significant between the two sub-categories (Dunn’s test with Bonferonni correction). ^3^ Nutrient adequacy ratio of recommended intake for children aged 4–13 years attending primary school [19], expressed per 100 kcal of dish. ^4^ Not significantly different from 5%/100 kcal. ^5^ All nutrient adequacy ratios significantly lower than 5%/100 kcal are shown in **
bold**. ^5^ One-sided test: significantly higher than 5%.

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
