# Peer review of "Nutritional Quality of Vegetarian and Non-Vegetarian Dishes at School: Are Nutrient Profiling Systems Sufficiently Informative?"

_nutrients, 2020, doi:10.3390/nu12082256_

Round 1

Reviewer 1 Report

This is a very good study on  the nutritional quality of vegetarian and non-vegetarian dishes offered at school. This is the first time that the nutritional quality of vegetarian dishes in school was assessed for a wide spectrum of dishes. The criticism about the nutrient profiling systems  provides a basis of consideration of the adequacy of the dishes served at school daily. This is a well balanced paper, with a systematic approach. I have some concerns that I noted below.  

  1. Title: I suggest a more explanatory title: g. “Vegetarian and non vegetarian dishes at school: are the nutrient profiling systems sufficient to represent their nutritional quality?”
  2. Line 17: ..being sub-divided into 3 sub-categories.. Suggest: has been divided into 3 subcategories.
  3. Line 39: about “ fifteen frequency criteria” : (e.g. side... out of 20) It is not clear the definition of 15 frequency criteria even in brackets.
  4. Line 49: I think that the definition of bioavailability is not necessary ( defined...pathways).
  5. Line 55: “Therefore… effects”.Suggest “ Seafood consumption should  be guaranteed to ensure adequate omega 3-fatty ...iodine intake but has to be limited to minimize possible detrimental health effects”
  6. Discussion:In the limitations of the study I would include that the sample is not homogeneous since obtained by merging database of 2015 and a database of 2019, which may be different to some extent.
  7. Line 369: Suggest to transform direct speech in indirect speech with a summary of position paper.
  8. Line 380: Suggest: Put the sentence “Potential …ADA” after “necessary” of line 376 as: … “necessary. Nonetheless, potential associations between..”
  9. Line 385: I suggest to cite the article: “ Baldassarre M.E. ; Panza R;Farella I. et al.  “Vegetarian and vegan weaning of the infant : how common and how evidence- based? A population-based survey and narrative review.” Int. J. Environ.Res. Public health 2020, 17, 4835. DOI: 10.3390/ijerph17134835.
  10. Line 431 to 434: I suggest replacing with:The present study demonstrates  that nutrient profiling systems alghoritms are not sufficient to determine adequacy of school meals, since they do not cover all “protective” nutrients. It is therefore advisable to maintain a critical approach towards these algorithms with the aim to ensure the correct intake of nutritional elements needed by the growing children.

Reviewer 2 Report

Interesting topic, but poorly explained. It is not clear what age the children are, whether there is a difference in portion size by age, and some of the measures may not be equally relevant to all children (e.g. energy density). The meals seem to be planned by a dietitian and cooked on site (?), so I am not sure why the one vegetarian meal per week should be an issue.

Interesting topic for the study considering the recent change to the requirements for school lunches to include a vegetarian option. I believe this is a positive move by the French government to make children aware of meat-free options.

Overall – difficult to follow in places, partially due long sentences and lack of correspondence to school meals in many other countries. More detailed explanation and simpler sentence structures would help improve the manuscript.

You seem to be working with averages of averages throughout the analysis. I am wondering whether it would be more useful for practice to analyse actual meals/meal combinations and provide relevant nutrient profiles for those.

Energy density of foods may not be a negative factor for children to the same extent as for adults as their gastric capacity is smaller than that of adults. Thus, Nutri-Score may not be the most suitable measure used. It is not clear what age group the children are. If there is a wide age range, is there a difference in portion size? Also, is the 2000 kcal average energy intake suitable for all children in the schools?

Specific comments:

Unit use – consider using the standard SI unit of kJ instead of the outdated kcal.

Abstract and intro – slightly confusing as cafeterias work differently in different countries – children purchase food from a range like in a café, and only if they want to as they can bring food from home. It sounds like there is a set menu in France that all children eat?

Abstract – explain vegetarian meal categories earlier if providing outcomes by categories in abstract.

Is there a generally accepted categorisation of nutrients into a “harmful” category? I have not come across this term before and I do not find it specifically helpful, as nutrients classed a ‘harmful’ in this manuscript will be present in a variety of foods in different amounts, and it is the foods we consume, not nutrients.

  1. 47-51 sentence to long, needs to be broken up by meaning to improve parsing

ll.89-90 – protein dish with side dish example makes no sense – green beans with steak – are green beans the protein here and the steak the side dish?

  1. 112 and following – I am not sure what is the purpose of creating an ‘average’ side dish
  2. 118 and following – explanation of combinations confusing and difficult to follow. Consider presenting this information in a table or using visual means
  3. 135 and following – present this information in a table, summarise main points in text

Also, I have two overarching conceptual questions:

Why did you expect the meals provided at the school not to be adequate? It sounds like they are planned by a dietician and cooked fresh, rather than consisting of processed convenience foods.

If there is one vegetarian meal per week, does it make a difference to the nutrition of children if that one meal per week is slightly lower in some nutrients?
